EMBO
Molecular Medicine

# Nuclear receptor corepressor 1 represses cardiac hypertrophy

Chao Li[1,2,3,†] (iD), Xue-Nan Sun[1,2,4,†], Bo-Yan Chen[1,2,†], Meng-Ru Zeng[1,2,4,†], Lin-Juan Du[1,2,4], Ting Liu[1,2], Hui-Hui Gu[5], Yuan Liu[1,2,4], Yu-Lin Li[1,2], Lu-Jun Zhou[1,2], Xiao-Jun Zheng[1,2,4], Yu-Yao Zhang[1,2,4], Wu-Chang Zhang[1,2], Yan Liu[1,2], Chaoji Shi[1,2], Shuai Shao[6], Xue-Rui Shi[7], Yi Yi[7], Xu Liu[7], Jun Wang[5], Johan Auwerx[8] (iD), Zhao V Wang[3], Feng Jia[6,*] (iD), Ruo-Gu Li[7,**] (iD) & Sheng-Zhong Duan[1,2,***] (iD)

## Abstract

The function of nuclear receptor corepressor 1 (NCoR1) in cardiomyocytes is unclear, and its physiological and pathological implications are unknown. Here, we found that cardiomyocyte-specific NCoR1 knockout (CMNKO) mice manifested cardiac hypertrophy at baseline and had more severe cardiac hypertrophy and dysfunction after pressure overload. Knockdown of NCoR1 exacerbated whereas overexpression mitigated phenylephrine-induced cardiomyocyte hypertrophy. Mechanistic studies revealed that myocyte enhancer factor 2a (MEF2a) and MEF2d mediated the effects of NCoR1 on cardiomyocyte hypertrophy. The receptor interaction domains (RIDs) of NCoR1 interacted with MEF2a to repress its transcriptional activity. Furthermore, NCoR1 formed a complex with MEF2a and class IIa histone deacetylases (HDACs) to suppress hypertrophy-related genes. Finally, overexpression of RIDs of NCoR1 in the heart attenuated cardiac hypertrophy and dysfunction induced by pressure overload. In conclusion, NCoR1 cooperates with MEF2 and HDACs to repress cardiac hypertrophy. Targeting NCoR1 and the MEF2/HDACs complex may be an attractive therapeutic strategy to tackle pathological cardiac hypertrophy.

**Keywords** cardiac hypertrophy; class IIa HDACs; MEF2a; nuclear receptor corepressor 1
**Subject Category** Cardiovascular System
See also: **A Grund & J Heineke** (November 2019)

## Introduction

Prolonged cardiac hypertrophy is an independent risk factor for heart failure (Frey & Olson, 2003), a major cause of morbidity and mortality worldwide (Benjamin *et al*, 2017). Pathological conditions such as long-term hypertension, myocardial infarction, and valvular stenosis often lead to heart failure (Heineke & Molkentin, 2006). Cardiac hypertrophy is a common theme during the deleterious development of these diseases (Heineke & Molkentin, 2006). Therefore, identifying efficient targets and strategies to restrain pathological cardiac hypertrophy remains to be a promising approach to curb the rising trend of heart failure.

Transcriptional regulation exerts fundamental roles in the process of pathological cardiac hypertrophy and heart failure (Heineke & Molkentin, 2006). Reactivation of fetal genes, including *Acta1*, *Nppa*, and *Nppb*, is a characteristic of cardiac hypertrophy and an important molecular mechanism underlying the hypertrophy of cardiomyocytes (Kuwahara *et al*, 2012). Transcription factors are critical players in regulating the upregulation of fetal genes and therefore promising targets for the intervention of cardiac hypertrophy (Heineke & Molkentin, 2006; Kohli *et al*, 2011). Among them, myocyte enhancer factor 2 (MEF2) family, including MEF2a, MEF2b, MEF2c, and MEF2d, has been considered as core transcription factors in cardiac development and reprogramming (Potthoff & Olson, 2007; Desjardins & Naya, 2016). Furthermore, MEF2a, MEF2c, and MEF2d are well characterized for their crucial roles in cardiac hypertrophy (van Oort *et al*, 2006; Xu *et al*, 2006; Kim *et al*, 2008; Gao *et al*, 2016), and MEF2b has been implicated to play a role in cardiac development (Molkentin *et al*, 1996).

1  Laboratory of Oral Microbiota and Systemic Diseases, Shanghai Ninth People's Hospital, College of Stomatology, Shanghai Jiao Tong University School of Medicine, Shanghai, China
2  National Clinical Research Center for Oral Diseases, Shanghai Key Laboratory of Stomatology & Shanghai Research Institute of Stomatology, Shanghai, China
3  Division of Cardiology, Department of Internal Medicine, University of Texas Southwestern Medical Center, Dallas, TX, USA
4  Shanghai Institute of Nutrition and Health, Shanghai Institutes for Biological Sciences, University of Chinese Academy of Sciences, Chinese Academy of Sciences, Shanghai, China
5  Shanghai Jing'an District Central Hospital, Fudan University, Shanghai, China
6  Department of Neurosurgery, Ren Ji Hospital, Shanghai Jiao Tong University School of Medicine, Shanghai, China
7  Department of Cardiology, Shanghai Chest Hospital, Shanghai Jiao Tong University, Shanghai, China
8  Laboratory of Integrative and Systems Physiology, Institute of Bioengineering, École Polytechnique Fédérale de Lausanne (EPFL), Lausanne, Switzerland
   *Corresponding author. Tel: +86 21 68383708; Fax: +86 21 58394262; E-mail: projiafeng@163.com
   **Corresponding author. Tel: +86 18017320201; Fax: +86 21 62803712; E-mail: 13564565961@163.com
   ***Corresponding author. Tel: +86 21 38452653; Fax: +86 21 63136856; E-mail: duansz@shsmu.edu.cn
   †These authors contributed equally to this work

Another critical aspect of gene transcriptional regulation is epigenetic modification, which is controlled by chromatin-modifying enzymes (Dirkx et al, 2013). Histone deacetylases (HDACs) are a group of such enzymes (Seto & Yoshida, 2014). Class IIa HDACs, containing HDAC4, HDAC5, HDAC7, and HDAC9, play protective roles during cardiac hypertrophy (McKinsey, 2011). Overexpression of HDAC4 (Backs et al, 2011), HDAC5 (Vega et al, 2004), or HDAC9 (Zhang et al, 2002) in neonatal rat ventricular myocytes (NRVMs) represses hypertrophy. Conversely, HDAC5 or HDAC9 knockout mice develop more severe cardiac hypertrophy after pressure overload (POL; Zhang et al, 2002; Chang et al, 2004). Moreover, class IIa HDACs are able to interact with MEF2 to regulate the transcription of fetal genes (Dirkx et al, 2013).

Transcriptional coregulators (including coactivators and corepressors) act as intermediates connecting to both transcription factors and chromatin-modifying enzymes, all of which together determine the final transcriptional output (Mottis et al, 2013). Therefore, these coregulators may play important roles in regulating hypertrophy of cardiomyocytes. It has been demonstrated that the coactivator p300 promotes cardiac hypertrophy and its deletion attenuates POL-induced hypertrophy (Slepak et al, 2001; Gusterson et al, 2003; Wei et al, 2008). However, less attention has been paid to the functions of corepressors during cardiac hypertrophy. Nuclear receptor corepressor 1 (NCoR1) and its homologous protein silencing mediator of retinoic acid and thyroid hormone receptor (SMRT) are among the most studied corepressors (Mottis et al, 2013). SMRT has been reported to participant in cardiac development (Jepsen et al, 2008). Although human studies have identified NCoR1 as a candidate gene for pathogenesis of left-sided congenital heart disease (Hitz et al, 2012), the exact functions of NCoR1 during cardiac development and disease have not been explored.

Here, we delineated the function of NCoR1 in cardiomyocytes both in vivo and in vitro, and explored the underlying mechanisms. We generated cardiomyocyte-specific NCoR1 knockout (CMNKO) mice and investigated the impact of cardiomyocyte NCoR1 deficiency under physiological and pathological condition. We then utilized NRVMs to study the roles of NCoR1 in phenylephrine-induced cardiomyocyte hypertrophy. Subsequently, we explored the underlying mechanisms how NCoR1 interacted with MEF2 and HDAC to regulate cardiac hypertrophy. Finally, we explored the possibility of gene therapy using NCoR1 for pathological cardiac hypertrophy.

# Results

### NCoR1 deficiency in cardiomyocytes leads to cardiac hypertrophy in mice

To explore the function of NCoR1 in cardiac hypertrophy, we first detected the expression of NCoR1 in hypertrophied hearts or cardiomyocytes. Heart samples from hypertrophic cardiomyopathy patients exhibited elevated expression of genes related to pathological cardiac hypertrophy, such as Nppa, Nppb, and Myh7 (Fig EV1A). Western blotting analysis revealed that the protein level of NCoR1 was elevated in heart samples from hypertrophic cardiomyopathy patients (Fig EV1B and C). In mouse model of cardiac hypertrophy induced by angiotensin II or abdominal aortic constriction (AAC),

the protein level of NCoR1 was also upregulated in hypertrophied hearts (Fig EV1D–I). In cultured NRVMs, phenylephrine increased the expression of NCoR1 (Fig EV1J). These results indicated that NCoR1 played an important role in cardiac hypertrophy.

We then generated cardiomyocyte-specific NCoR1 knockout (CMNKO) mice by crossing floxed NCoR1 mice (Yamamoto et al, 2011) with αMHC-Cre mice (Duan et al, 2005; Appendix Fig S1A). Western blotting and qRT–PCR analysis revealed that NCoR1 was specifically deleted in heart samples of CMNKO mice (Appendix Fig S1B and C).

Unexpectedly, CMNKO mice manifested cardiac hypertrophy at baseline as demonstrated by the ratios of ventricular weight to tibia length (VW/TL) and ventricular weight to body weight (VW/BW), particularly at 10 months old (Fig 1A and B). H&E and wheat germ agglutinin staining illustrated that cross-sectional areas of cardiomyocytes in CMNKO mice were significantly larger (Fig 1C–E). Expression of fetal genes, including Acta1, Nppa, and Nppb, was significantly upregulated in heart samples of CMNKO mice (Fig 1F). Furthermore, both ejection fraction (EF) and fractional shortening (FS) were significantly lower in CMNKO mice at 10 months old, indicating impaired cardiac function (Fig 1G and H, Appendix Table S1). Ten-month-old αMHC-Cre mice had similar cardiac function comparing to age-matched LC mice (Fig EV2 and Appendix Table S2), indicating that the αMHC-Cre mice we used did not cause cardiac toxicity at this age.

Blood pressures were comparable between CMNKO and LC mice (Fig EV3A), excluding the influence of blood pressure difference on cardiac hypertrophy. Picrosirius red staining showed no difference in cardiac fibrosis between CMNKO and LC mice, although cardiac expression of fibrosis-related genes was augmented in CMNKO mice (Fig EV3B and C).

### NCoR1 deficiency in cardiomyocytes exacerbates abdominal aortic constriction-induced cardiac hypertrophy

Next, we subjected 8- to 12-week-old LC and CMNKO mice to AAC and investigated the impacts of NCoR1 deficiency on pathological cardiac hypertrophy. VW/BW was significantly increased in CMNKO mice compared to LC mice 2 weeks after AAC (Fig 2A and B). H&E and WGA staining illustrated marked augmentation of cardiomyocyte size in CMNKO mice after AAC (Fig 2C–E). Consistently, expression of fetal genes, such as Acta1, Nppa, and Nppb, was significantly increased in ventricular samples of CMNKO mice after AAC (Fig 2F). Picrosirius red staining showed more severe cardiac fibrosis in ventricular samples of CMNKO mice after AAC (Fig 2G and H). Accordingly, expression of fibrosis-related genes, such as Col1a1, Col1a2, CTGF, and Fn1, was also upregulated in ventricular samples of CMNKO mice compared to LC mice after AAC (Fig 2I).

Long-term pressure overload leads to cardiac dysfunction (Heineke & Molkentin, 2006). Echocardiograph analysis showed that both ejection fraction and fractional shortening were more severely impaired in CMNKO mice than LC mice after AAC (Fig 2J and K, Appendix Table S3). Furthermore, the ratio of lung weight to body weight (LW/BW) was remarkably increased in CMNKO mice compared to LC mice after AAC (Fig 2L), indicating more severe heart failure in CMNKO mice. The survival rate of CMNKO mice was markedly lower than that of LC mice after AAC (Fig 2M). These

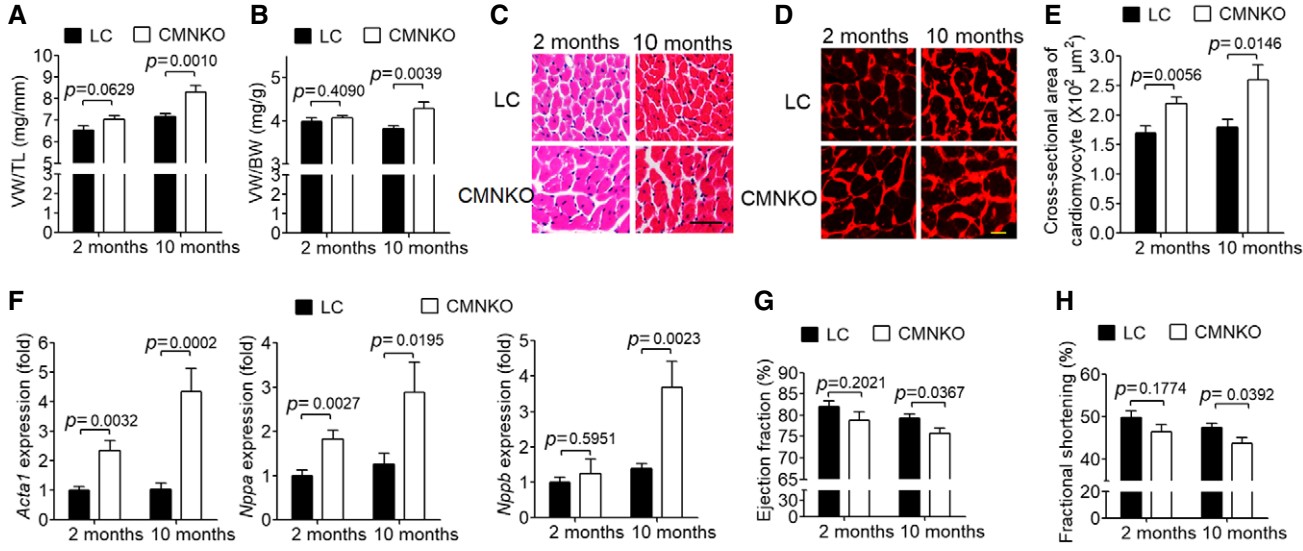

**Figure 1. Deficiency of cardiomyocyte NCoR1 leads to cardiac hypertrophy and dysfunction.**

A     Ventricular weight-to-tibia length ratio (VW/TL) of littermate control (LC) or cardiomyocyte-specific NCoR1 knockout (CMNKO) mice. *n* = 10:9:12:7.
B     Ventricular weight-to-body weight ratio (VW/BW) of LC or CMNKO mice. *n* = 10:8:12:7.
C     Representative H&E staining of cross sections of left ventricles. Scale bar: 50 μm.
D     Representative wheat germ agglutinin (WGA) staining of cross section of left ventricles. Scale bar: 10 μm.
E     Quantification of cardiomyocyte size. *n* = 11:10:8:8.
F     qRT–PCR analysis of hypertrophy-related genes in left ventricles. *n* = 9:9:11:8.
G, H  (G) Ejection fraction and (H) fractional shortening measured by echocardiography. *n* = 9:7:14:10.

Data information: Data are presented as mean ± SEM. Student's *t*-test was used for statistical analysis.

results together demonstrated a protective role of NCoR1 in pathological cardiac hypertrophy.

**NCoR1 inhibits phenylephrine (PE)-induced cardiomyocyte hypertrophy**

We then investigated the role of NCoR1 in cardiomyocyte hypertrophy using cultured NRVMs. NCoR1 knockdown using siRNA (Fig 3A) significantly increased surface area of NRVMs under the stimulation of PE (Fig 3B and C). Consistently, expression of fetal genes, including *Acta1* and *Nppa*, was upregulated in NRVMs by NCoR1 knockdown under PE stimulation (Fig 3D). Conversely, overexpression of NCoR1 in NRVMs using lentivirus (Fig 3E) strikingly decreased surface area of NRVMs under PE stimulation (Fig 3F and G). Expression of *Acta1* and *Nppa* was also markedly inhibited by NCoR1 overexpression (Fig 3H). Together, these results demonstrated that NCoR1 was a negative regulator for hypertrophy of cardiomyocytes *in vitro*.

**MEF2a and MEF2d mediate the suppressive effects of NCoR1 on cardiomyocyte hypertrophy**

We next asked how NCoR1 regulated hypertrophy of cardiomyocytes. Previous studies have revealed that NCoR1 represses a variety of transcription factors (Mottis *et al*, 2013). MEF2a, MEF2c, and MEF2d have been demonstrated to play vital roles in cardiac hypertrophy (Xu *et al*, 2006; Kim *et al*, 2008). Results of RNA-sequencing (Dataset EV1) demonstrated that 57 genes were most significantly (fold change > 2, FDR < 0.05) upregulated in heart samples from

CMNKO mice compared to LC mice (Appendix Fig S2A). Eleven of these 57 genes are regulated by MEF2 (Appendix Fig S2B). Some of these 11 genes were validated by qRT–PCR in ventricular samples (Figs 1F and EV3C). We also detected the expressions of cJun, monocyte chemoattractant protein 1, and glucose transporter type 4, which had been reported to be target genes of MEF2 (Han *et al*, 1997; Suzuki *et al*, 2004; Amoasii *et al*, 2016). As expected, they were all upregulated in ventricular samples from CMNKO mice (Appendix Fig S2C).

Therefore, we explored the possibility that MEF2 mediated the effects of NCoR1 in cardiomyocytes. Under PE stimulation, MEF2a knockdown using siRNA (Fig EV4A) almost completely abolished the hypertrophic effects of NCoR1 knockdown in NRVMs (Fig 4A and B). Similarly, MEF2d knockdown (Fig EV4A) blocked the hypertrophic effects of NCoR1 knockdown (Fig 4C and D). Consistently, the upregulated expression of *Acta1* and *Nppa* in NRVMs by NCoR1 knockdown was markedly attenuated after MEF2a or MEF2d knockdown (Fig 4E). However, MEF2c knockdown did not affect the hypertrophic effects of NCoR1 knockdown (Fig EV4B and C). Collectively, these results indicated that NCoR1 inhibited the expression of fetal genes and cardiomyocyte hypertrophy through repressing the activities of MEF2a and/or MEF2d.

**NCoR1 directly interacts with MEF2a to suppress its transcriptional activity**

MEF2a and MEF2d belong to the same family and share similar gene architecture (Potthoff & Olson, 2007). We went further to explore the molecular mechanisms by which NCoR1 regulated the activity

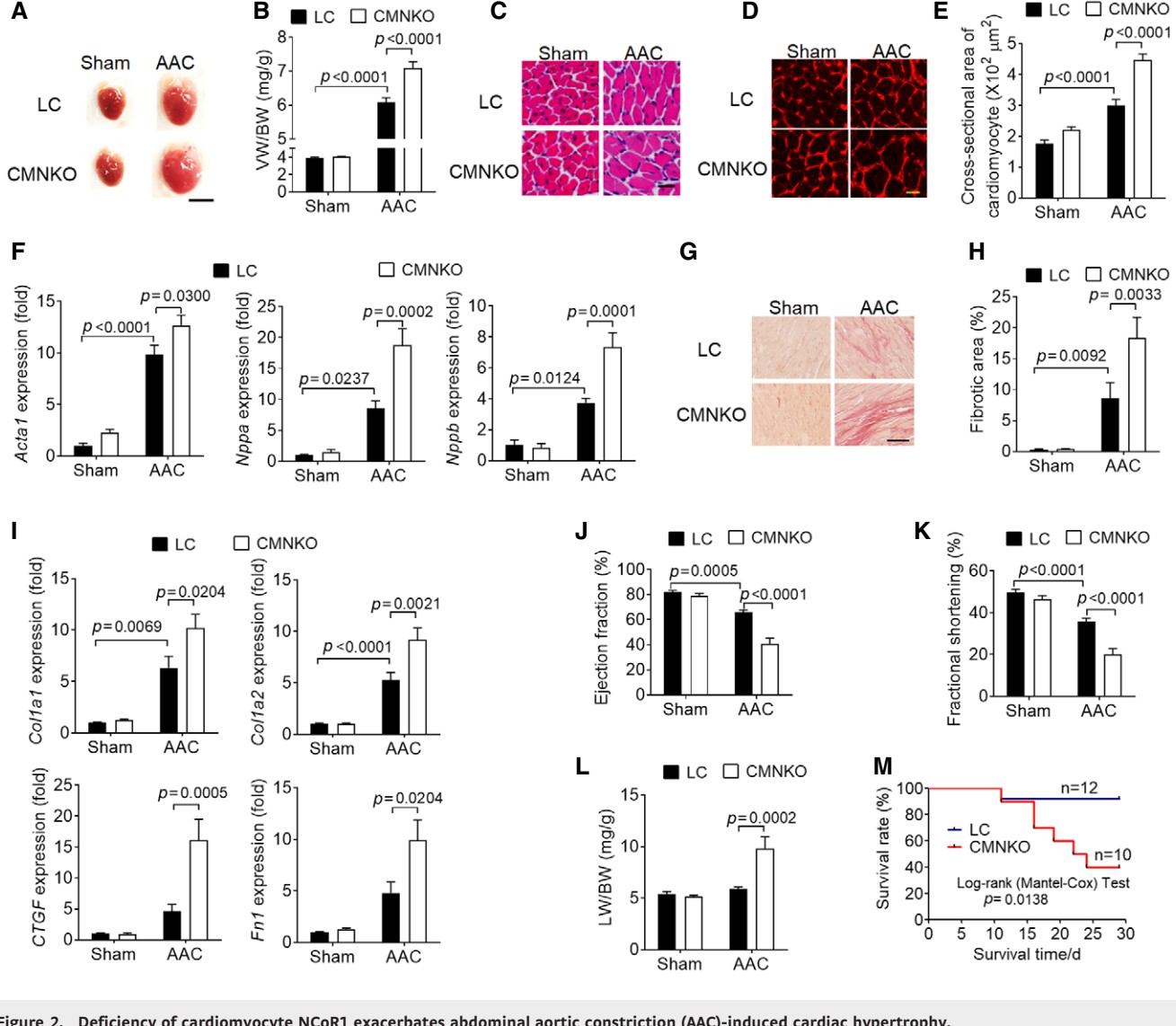

**Figure 2.  Deficiency of cardiomyocyte NCoR1 exacerbates abdominal aortic constriction (AAC)-induced cardiac hypertrophy.**

A  Representative images of hearts from LC or CMNKO mice subjected to sham operation or AAC for 2 weeks. Scale bar: 0.5 cm.
B  VW/BW of LC or CMNKO mice 2 weeks after sham operation or AAC. *n* = 9:9:16:15.
C  Representative H&E staining of cross sections of left ventricles. Scale bar: 25 μm.
D  Representative WGA staining. Scale bar: 10 μm.
E  Quantification of cardiomyocyte size. *n* = 10:10:8:9.
F  qRT–PCR analysis of hypertrophy-related genes in left ventricles. *n* = 8:7:15:15.
G  Representative picrosirius red staining of cross sections of left ventricles. Scale bar: 50 μm.
H  Quantification of fibrotic areas. *n* = 11:10:10:9.
I  qRT–PCR analysis of fibrosis-related genes in left ventricles. *n* = 8:7:15:15.
J, K  (J) Ejection fraction and (K) fractional shortening measured by echocardiography. *n* = 9:7:14:12.
L  Lung weight-to-body weight ratio (LW/BW) of LC or CMNKO mice 2 weeks after sham operation or AAC. *n* = 8:7:16:15.
M  Cumulative survival rate of LC or CMNKO mice subjected to AAC. *n* = 12:10.

Data information: Data are presented as mean ± SEM. Two-way ANOVA followed by Bonferroni post-tests was used for statistical analysis.

of MEF2a. Luciferase assays demonstrated that MEF2a significantly induced the transcriptional activity of both *Acta1* promoter and *Nppa* promoter and that NCoR1 markedly suppressed such induction (Fig 5A). Co-IP analysis showed that NCoR1 interacted with MEF2a (Fig 5B). Co-IP analysis using truncated MEF2a demonstrated that the DNA-binding domain but not the transcriptional activation domain interacted with NCoR1 (Fig 5C). Co-IP analysis using truncated NCoR1 revealed that receptor interaction domains (RIDs) of NCoR1 interacted with MEF2a (Fig 5D). Interestingly, transfection of RIDs (1939-2453) was sufficient to repress MEF2a-induced elevation of transcriptional activities of *Acta1* and *Nppa* promoters (Fig 5E). Moreover, overexpression of RIDs (1939-2453) in NRVMs (Fig 5F) was enough to inhibit PE-induced increase of surface area and fetal gene expression (Fig 5G–I).

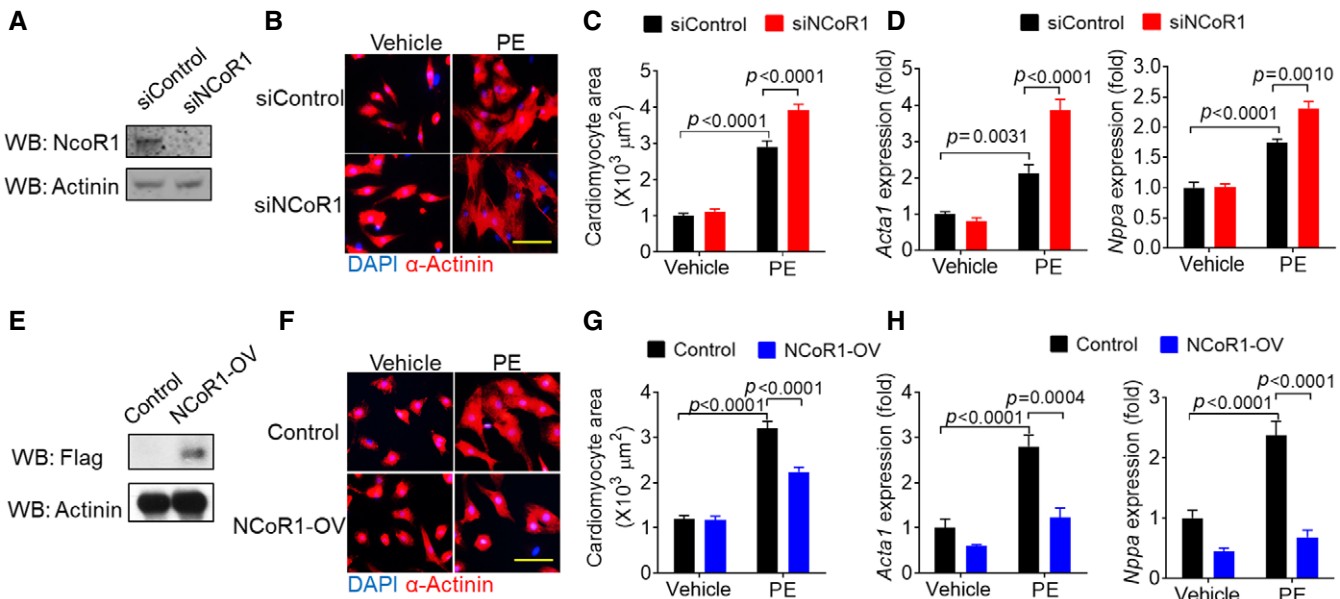

**Figure 3. NCoR1 inhibits phenylephrine-induced cardiomyocyte hypertrophy.**

A   Western blotting analysis of NCoR1 in neonatal rat ventricular myocytes (NRVMs) transfected with siControl or siNCoR1 for 3 days. siControl indicates control siRNA; siNCoR1, NCoR1 siRNA.

B   Representative immunofluorescence staining of α-Actinin in NRVMs transfected with siRNA for 48 h and then treated with vehicle (H₂O) or phenylephrine (PE) for another 48 h. Scale bar: 50 μm.

C   Quantification of the surface area of α-Actinin-positive NRVMs with or without NCoR1 knockdown. A total of 40 NRVMs were randomly chosen from four replicate coverslips for each group and used for statistical analysis.

D   qRT–PCR analysis of Acta1 and Nppa in NRVMs with or without NCoR1 knockdown. *n* = 4:4.

E   Western blotting analysis of NCoR1-flag in NRVMs infected by control lentivirus (Control) or NCoR1-flag lentivirus (NCoR1-OV) for 4 days.

F   Representative immunofluorescence staining of α-Actinin in NRVMs infected with lentivirus for 48 h and then treated with vehicle or PE for another 48 h. Scale bar: 50 μm.

G   Quantification of the surface area of α-Actinin-positive NRVMs with or without NCoR1 overexpression. A total of 30 NRVMs were randomly chosen from three replicate coverslips for each group and used for statistical analysis.

H   qRT–PCR analysis of Acta1 and Nppa in NRVMs with or without NCoR1 overexpression. *n* = 4:3.

Data information: Data represent three independent experiments. Data are presented as mean ± SEM. Two-way ANOVA followed by Bonferroni post-tests was used for statistical analysis.

Source data are available online for this figure.

## NCoR1 forms a complex with MEF2 and Class IIa HDACs to repress the expression of fetal genes *in vivo*

To explore the mechanisms how NCoR1 suppressed the expression of hypertrophy-related genes *in vivo*, we performed ChIP experiments using ventricular samples. ChIP results using antibodies against NCoR1 uncovered that NCoR1 was enriched within MEF2-binding regions but not remote regions on promoters of *Acta1* and *Nppa* (Fig 6A), suggesting that NCoR1 bound to MEF2 to regulate expression of *Acta1* and *Nppa in vivo*. ChIP results showed that more MEF2a was recruited to the MEF2-binding sites on *Acta1* and *Nppa* promoters in ventricular samples from CMNKO mice compared to those from LC mice (Fig 6B).

NCoR1 often recruits HDACs to execute its repressive function (Mottis *et al*, 2013). It has been widely reported that class IIa HDACs, including HDAC4, HDAC5, HDAC7, and HDAC9, interact with MEF2 to repress its transcriptional activities (Zhang *et al*, 2002). Therefore, we tried to determine whether NCoR1 would affect the function of HDACs. Results of immunofluorescence staining illustrated that NCoR1 overexpression inhibited the translocation of HDAC4 and HDAC5 from nucleus to cytosol in NRVMs after

PE treatment (Fig EV5A–D), indicating repression on their function (Clocchiatti *et al*, 2011). Conversely, ChIP results revealed that less HDAC4 and HDAC5 were recruited to the MEF2-binding sites on Acta1 and Nppa promoters in ventricular samples from CMNKO mice compared to those from LC mice (Fig 6C and D). As a result, acetylated-Histone 4 and acetylated-Histone 3 were increased on the promoters of *Acta1* and *Nppa* in ventricular samples from CMNKO mice (Fig 6E and F). Recruitment of more RNA polymerase II is an important characteristic of active gene expression (Sayed *et al*, 2013). ChIP results demonstrated that more RNA polymerase II was enriched in the transcriptional control regions of *Acta1* and *Nppa* in ventricular samples from CMNKO mice (Fig 6G).

To further understand how NCoR1 would affect the function of class IIa HDACs, we investigated whether NCoR1 directly interacted with class IIa HDACs. Co-IP experiments were performed after NCoR1 and class IIa HDACs (but not MEF2a) were overexpressed in NRVMs. We did not detect direct interactions between NCoR1 and HDAC4, HDAC5, or HDAC9 in NRVMs (Fig EV5E). These results indicated MEF2 was an important connecting protein between NCoR1 and class IIa HDACs in cardiomyocytes. This notion was further substantiated by results of Co-IP experiments using MEF2a

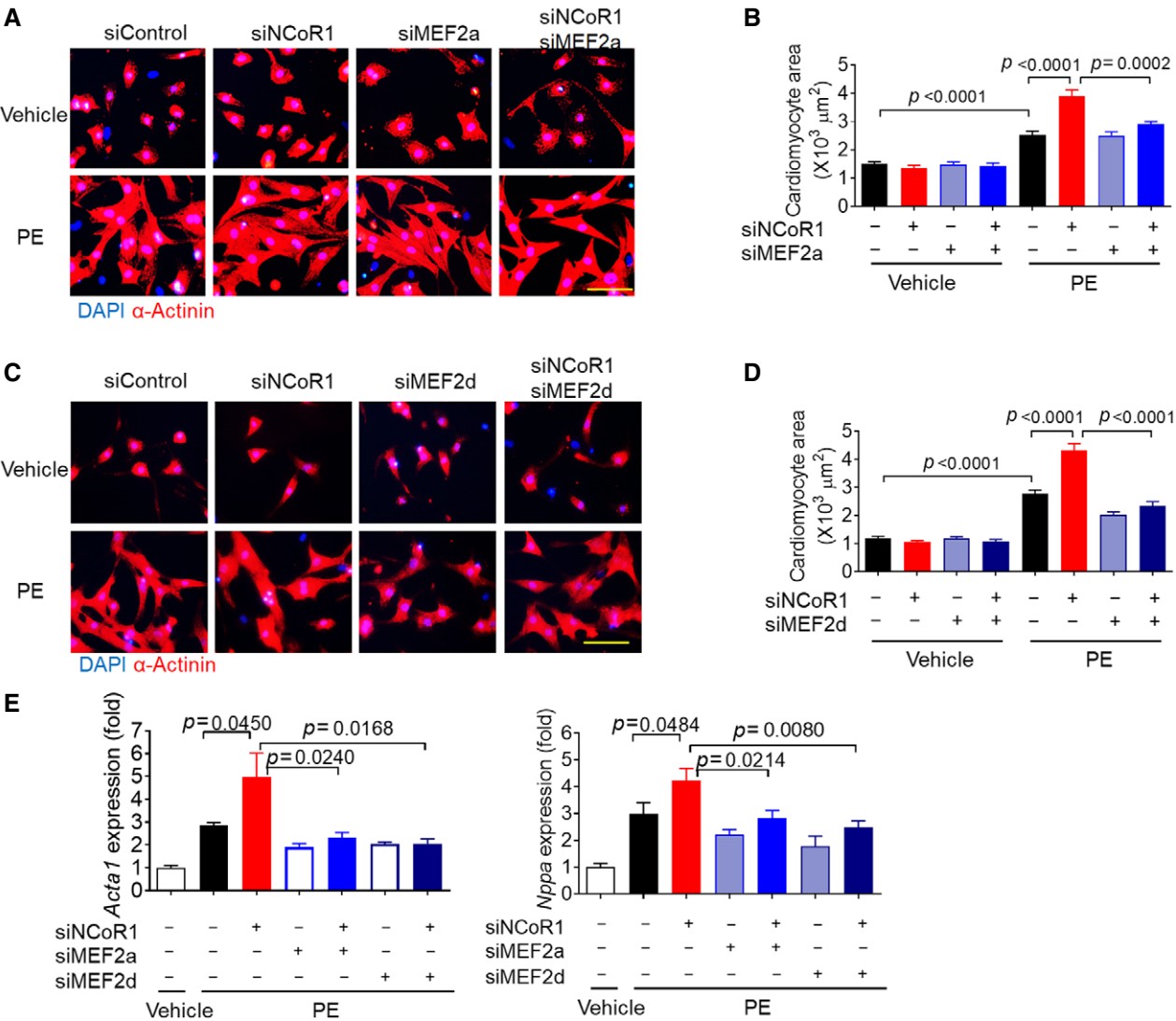

**Figure 4. MEF2a and MEF2d mediate the impacts of NCoR1 knockdown on PE-induced cardiomyocyte hypertrophy.**

A   Representative immunofluorescence staining of α-Actinin in NRVMs transfected with siRNA for 48 h and then treated with vehicle ($H_2O$) or PE for another 48 h. Scale bar: 50 μm. siMEF2a indicates MEF2a siRNA.
B   Quantification of the surface area of α-Actinin-positive NRVMs with or without knockdown of NCoR1 and/or MEF2a. A total of 15–20 NRVMs were randomly chosen from three replicate coverslips for each group and used for statistical analysis.
C   Representative immunofluorescence staining of α-Actinin in NRVMs transfected with siRNA for 48 h and then treated with vehicle or PE for another 48 h. Scale bar: 50 μm. siMEF2d indicates MEF2d siRNA.
D   Quantification of the surface area of α-Actinin-positive NRVMs with or without knockdown of NCoR1 and/or MEF2d. A total of 15–19 NRVMs were randomly chosen from three replicate coverslips for each group and used for statistical analysis.
E   qRT–PCR analysis of Acta1 and Nppa in NRVMs. $n = 4$.

Data information: Data represent three independent experiments. Data are presented as mean ± SEM. Student's $t$-test was used for statistical analysis.

antibodies for IP, which revealed interactions between MEF2a and NCoR1 as well as MEF2a and HDAC4 in ventricular samples (Fig EV5F). Further, when MEF2a was overexpressed together with NCoR1 and HDAC5, interactions between NCoR1 and MEF2a as well as NCoR1 and HDAC5 were observed in NRVMs (Fig EV5G).

Therefore, we propose a working model for the mechanism by which NCoR1 functions in cardiomyocytes. In normal cardiomyocytes, NCoR1 forms a complex with MEF2 and class IIa HDACs. This complex inhibits the transcriptional activity of MEF2 and deacetylates histones to repress gene expression. When NCoR1 is

absent, this complex is destructed, more MEF2a is recruited to its binding sites, histones are more acetylated, and more RNA polymerase II is recruited, leading to induction of fetal genes (Fig 6H).

HDAC3 was considered to be a main enzyme responsible for the repressive activity of NCoR1 (Mottis *et al*, 2013). Mice with liver-specific NCoR1 deletion recapitulated the phenotype of mice with liver-specific HDAC3 knockout (Sun *et al*, 2013). Cardiomyocyte-specific HDAC3 knockout leads to cardiac hypertrophy and imbalance of lipid disposal in mouse heart (Montgomery *et al*, 2008; Sun *et al*, 2011). We analyzed whether NCoR1 would affect lipid

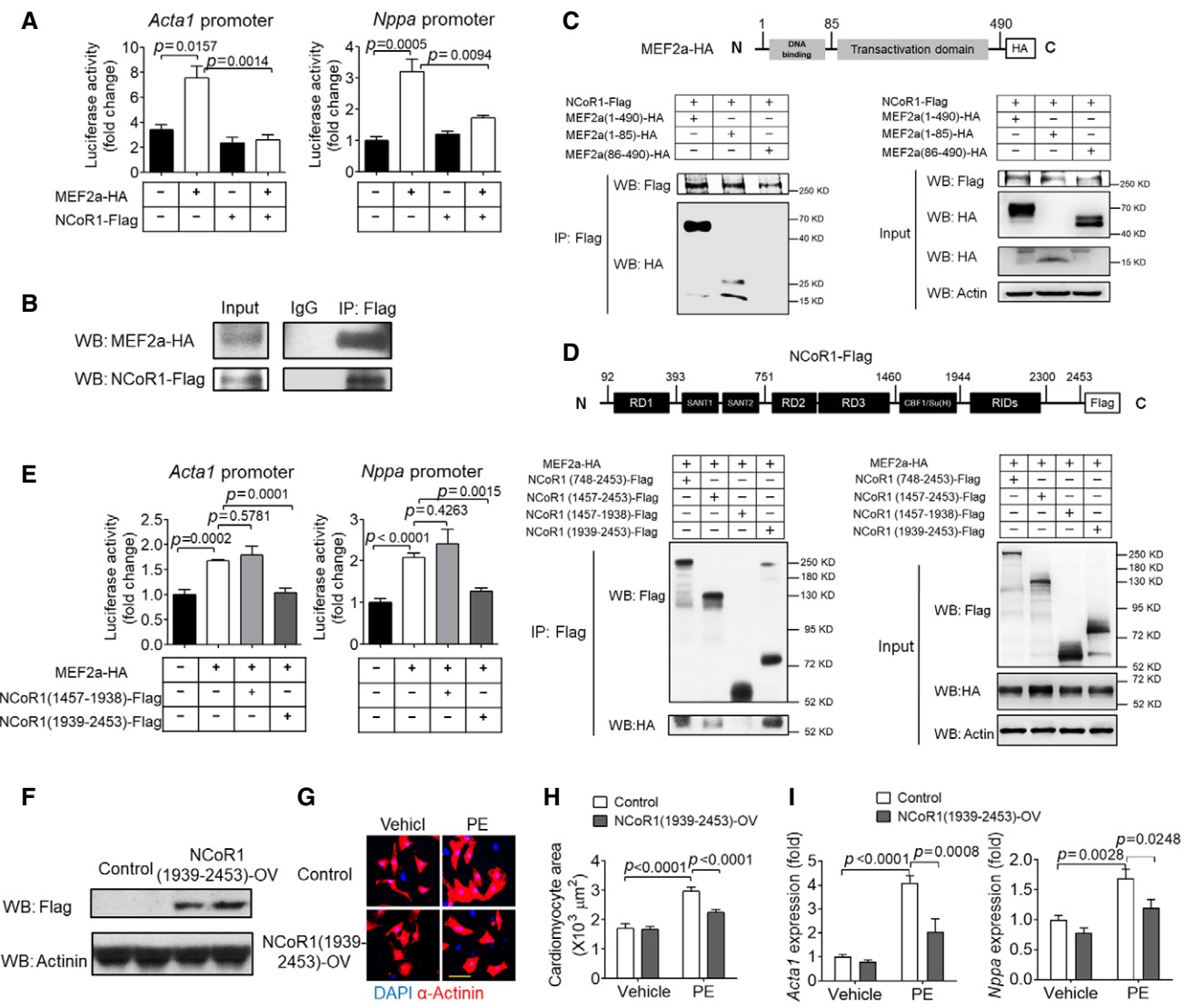

**Figure 5. NCoR1 directly interacts with MEF2a to suppress its transcriptional activity.**

A   Luciferase assays of Acta1 and Nppa promoters in HEK293FT cells transfected with NCoR1-Flag and/or MEF2a-HA or with empty plasmids. *n* = 3.
B   Co-immunoprecipitation (Co-IP) analysis of NCoR1 and MEF2a in HEK293FT cells transfected with full-length NCoR1-Flag and MEF2a-HA.
C   Co-IP analysis of NCoR1 and truncated MEF2a in HEK293FT cells. Schematic illustration of MEF2a-HA construct is shown above the Co-IP results.
D   Co-IP analysis of MEF2a and truncated NCoR1 in HEK293FT cells. Schematic illustration of NCoR1-Flag construct is shown above the Co-IP results.
E   Luciferase assays of Acta1 and Nppa promoters in HEK293FT cells transfected with full-length MEF2a and domain-specific NCoR1. *n* = 3.
F   Western blotting analysis of NCoR1-flag in NRVMs infected by control lentivirus (Control) or NCoR1 (1939-2453)-flag lentivirus [NCoR1(1939-2453)-OV] for 4 days.
G   Representative immunofluorescence staining of α-Actinin in NRVMs infected with lentivirus for 48 h and then treated with vehicle (DMSO) or PE for another 48 h. Scale bar: 50 μm.
H   Quantification of the surface area of α-Actinin-positive NRVMs. A total of 30 NRVMs were randomly chosen from three replicate coverslips for each group and used for statistical analysis.
I   qRT–PCR analysis of Acta1 and Nppa in NRVMs. *n* = 4.

Data information: Data represent three independent experiments. Data are presented as mean ± SEM. Student's *t*-test was used for statistical analysis in (A and E). Two-way ANOVA followed by Bonferroni post-tests was used for statistical analysis in (H and I).
Source data are available online for this figure.

metabolism in the heart. Triglyceride contents of left ventricles from LC and CMNKO mice were comparable under both fed and fasted conditions (Appendix Fig S3A). Oil red O staining of heart sections did not reveal difference between LC and CMNKO mice either (Appendix Fig S3B). These results implied that NCoR1 did not affect energy homeostasis in cardiomyocytes, differing from HDAC3.

**Overexpression of RIDs of NCoR1 in myocardium attenuated cardiac hypertrophy and dysfunction induced by AAC**

To apply our findings to the prospect of gene therapy, we tried to transfer *NCoR1* into myocardium by adeno-associated virus serotype 9 (AAV9) to examine its inhibitory effects on cardiac

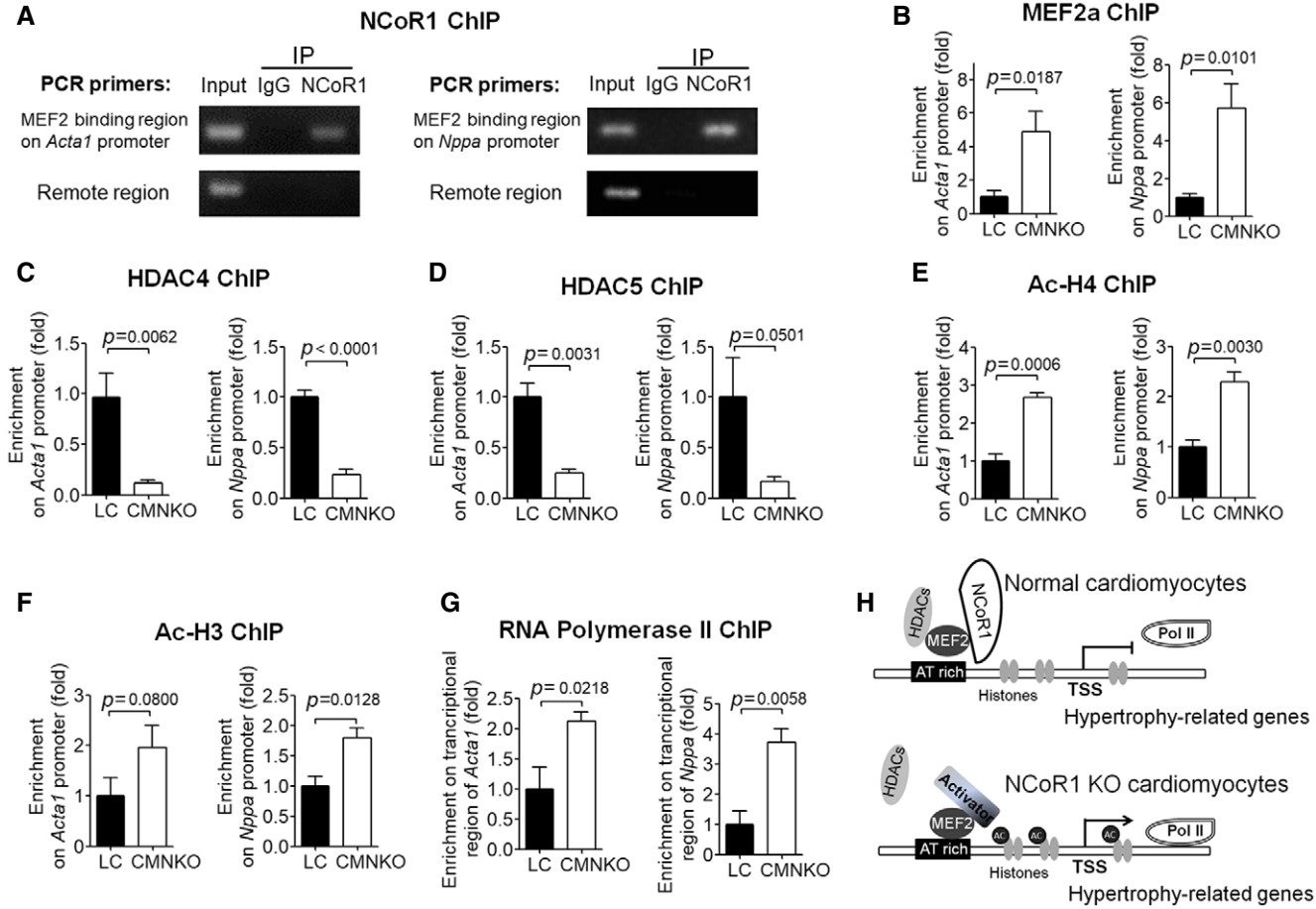

**Figure 6. NCoR1 forms a complex with MEF2 and HDAC4 to repress the expression of hypertrophy-related genes *in vivo*.**

A  ChIP analysis showing enrichment of NCoR1 in MEF2-binding regions on the promoters of *Acta1* and *Nppa* in ventricular samples.

B  ChIP analysis of enrichment of MEF2a on promoters of *Acta1* and *Nppa* in ventricular samples. *n* = 3:3.

C–F  ChIP analysis of enrichment of HDAC4 (C), HDAC5 (D), acetylated-Histone 4 (ac-H4) (E), and ac-H3 (F) on promoters of *Acta1* and *Nppa* in ventricular samples. *n* = 4:4 for (C), *n* = 3:3 for (D, E, F).

G  ChIP analysis of RNA polymerase II enrichment on the transcriptional regions of Acta1 and Nppa in ventricular samples. *n* = 3:3.

H  Working model of NCoR1 function in cardiomyocytes. TSS indicates transcription start site.

Data information: Data are presented as mean ± SEM. Student's *t*-test was used for statistical analysis.

hypertrophy. The full length of *NCoR1* (~7 kb) is too large to be inserted into AAV9 plasmid. Because RIDs of NCoR1 were able to inhibit hypertrophy in NRVMs (Fig 4G–I), we delivered the RIDs into mouse myocardium using AAV9 [AAV9 NCoR1 (1939-2453)-Flag]. Overexpression of RIDs (Fig 7A) significantly reduced VW/BW after AAC (Fig 7B) and decreased the size of cardiomyocytes as shown by H&E and WGA staining (Fig 7C–E). Accordingly, expression of fetal genes was suppressed in ventricular samples of mice infected with AAV9 NCoR1 (1939-2453)-Flag after AAC (Fig 7F). AAV9 NCoR1 (1939-2453)-Flag also inhibited cardiac fibrosis and expression of fibrosis-related genes after AAC (Fig 7G–I). In addition, cardiac dysfunction induced by AAC was significantly attenuated in mice infected with AAV9 NCoR1 (1939-2453)-Flag (Fig 7J and K, Appendix Table S4). These results demonstrated that overexpression of RIDs of NCoR1 in myocardium was sufficient to mitigate AAC-induced cardiac hypertrophy and dysfunction.

## Discussion

Although NCoR1 has been demonstrated to be an important player in transcriptional regulation in several other cell types, its function in cardiomyocytes and cardiac hypertrophy have not been elucidated. Through this study, we showed that cardiomyocyte NCoR1 deficiency led to cardiac hypertrophy under physiological condition and aggravated cardiac hypertrophy under POL. *In vitro* study further confirmed the suppressive role of NCoR1 in regulating the size of cardiomyocytes. Mechanistically, NCoR1 interacted with MEF2a and cooperated with class IIa HDACs to control cardiac hypertrophy. Importantly, overexpression of RIDs of NCoR1 in myocardium inhibited POL-induced cardiac hypertrophy and dysfunction.

NCoR1 may be considered as a stress-responsive and cardioprotective regulator during cardiac hypertrophy. Our results showed that the protein expression of NCoR1 was upregulated after

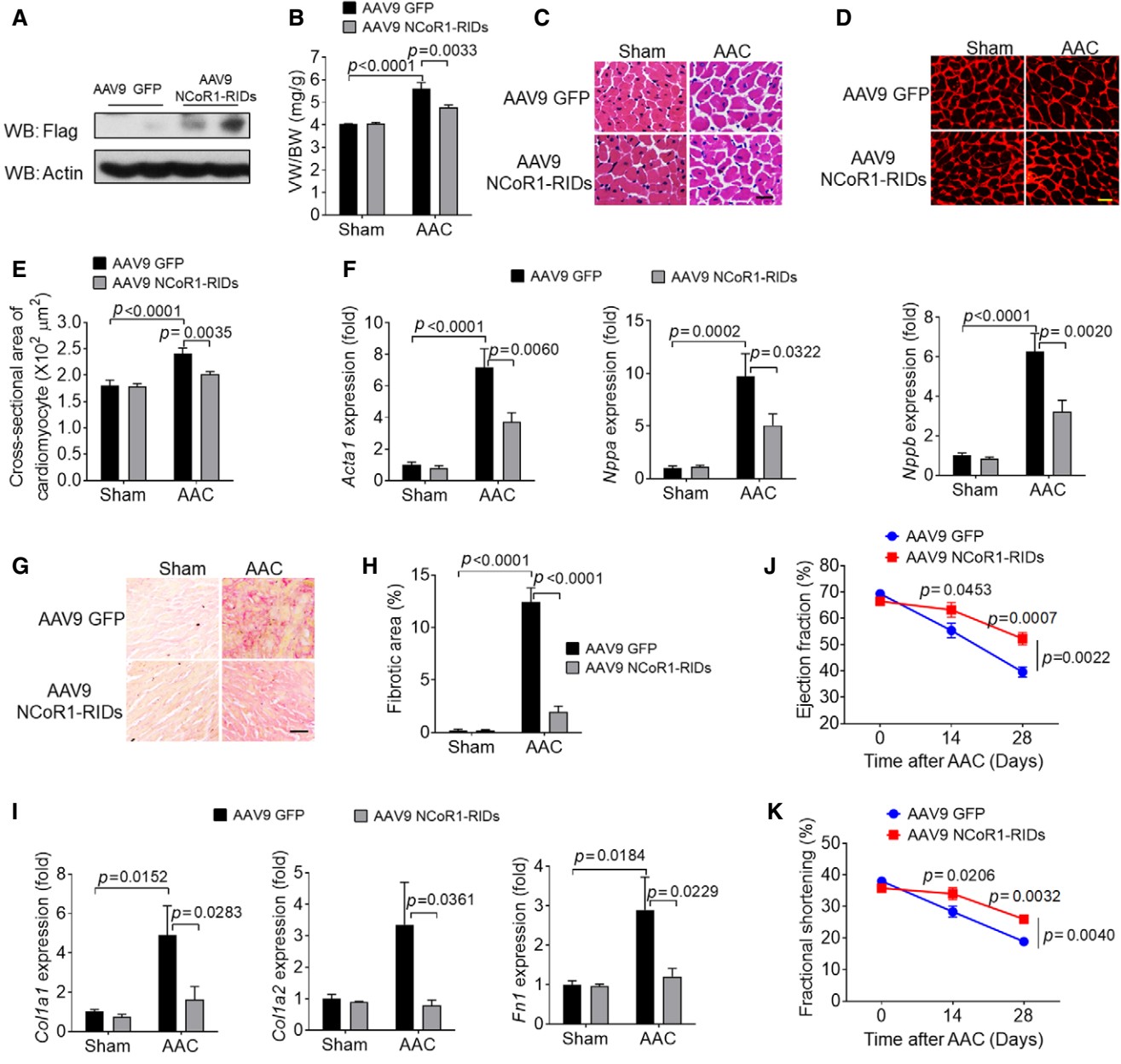

**Figure 7. Overexpression of RIDs of NCoR1 inhibits AAC-induced cardiac hypertrophy.**

A   Western blotting analysis of RIDs of NCoR1 in left ventricular samples from mice infected with AAV9 GFP or AAV9 NCoR1-RIDs (1939-2453)-Flag.
B   VW/BW of mice 2 weeks after sham operation or AAC. n = 7:7:12:12.
C   Representative H&E staining of cross sections of left ventricles. Scale bar: 10 μm.
D   Representative WGA staining. Scale bar: 10 μm.
E   Quantification of cardiomyocyte size. n = 6:6:7:8.
F   qRT–PCR analysis of hypertrophy-related genes in left ventricles. n = 8:8:11:11.
G   Representative picrosirius red staining of cross sections of left ventricles. Scale bar: 20 μm.
H   Quantification of fibrotic areas. n = 5:5:7:7.
I   qRT–PCR analysis of fibrosis-related genes in left ventricles. n = 8:8:11:11.
J, K   (J) Ejection fraction and (K) fractional shortening measured before, 2 and 4 weeks after AAC. n = 6:6.

Data information: Data are presented as mean ± SEM. Two-way ANOVA followed by Bonferroni post-tests was used for statistical analysis.
Source data are available online for this figure.

hypertrophic stimulation, which we believed to be part of the adaptation during cardiac hypertrophy. Indeed, NCoR1 in turn served as a suppressor of cardiac hypertrophy. Similarly, other regulators such as NGF1A-binding protein, inducible cAMP early repressor, and interferon regulatory factor 9 play inhibitory roles in cardiac hypertrophy, and the expression of these molecules all increases in

response to hypertrophic stimuli (Tomita *et al*, 2003; Hardt & Sadoshima, 2004; Buitrago *et al*, 2005; Jiang *et al*, 2014; Guo *et al*, 2018).

We presented strong evidence of interactions among NCoR1, MEF2, and class IIa HDACs to regulate cardiomyocyte size. NCoR1 usually functions through transcription factors (Mottis *et al*, 2013). Although previous studies have shown that NCoR1 deficiency increases the activity of MEF2d in skeletal muscle, it is unclear whether NCoR1 directly interacts with any member in the MEF2 family (Yamamoto *et al*, 2011). We identified MEF2a and MEF2d as key transcription factors that mediated the effects of NCoR1 in cardiomyocytes. Moreover, our data illustrated direct interactions between NCoR1 and MEF2a. We went further to map out the precise interaction domains, the RIDs in NCoR1 and the DNA-binding domain in MEF2a. Our data also indicated that MEF2a served as a critical connecting point between NCoR1 and class IIa HDACs. In addition, we uncovered the suppressive nature of NCoR1 on MEF2a and HDACs, the latter of which in turn regulated histone acetylation. These findings provided new insights into the mechanisms how the activities of MEF2a and HDACs were modulated, which might be useful for developing new strategies to intervene pathological cardiac hypertrophy.

NCoR1 usually cooperates with HDACs to execute its repressive activities (Mottis *et al*, 2013). In this study, we identified that class IIa HDACs were involved in the process in cardiomyocytes. HDAC3, a member of class I HDACs, has been previously demonstrated to participate in the repressive activities of NCoR1 (Mottis *et al*, 2013). For instance, similar phenotypes have been observed in liver-specific NCoR1 knockout and HDAC3 knockout mice, indicating the interaction between NCoR1 and HDAC3 in hepatocytes (Sun *et al*, 2013). HDAC3 deficiency in cardiomyocytes causes severe cardiac hypertrophy at early age and excessive accumulation of triglycerides, which is accompanied by upregulation of genes related to energy metabolism (Montgomery *et al*, 2008). In our study, NCoR1 deficiency in cardiomyocytes only led to mild cardiac hypertrophy phenotype under physiological condition and did not affect triglyceride accumulation in the heart. Therefore, NCoR1 works more likely through class IIa HDACs instead of HDAC3 to affect cardiac hypertrophy.

AAV9-mediated gene transfer was considered as a promising therapeutic route for heart disease (Pacak *et al*, 2006). Through exploration of the molecular mechanism, we identified that the RIDs was critical for NCoR1 to inhibit the transcriptional activity of MEF2a. We further demonstrated that overexpression of RIDs suppressed cardiomyocyte hypertrophy *in vitro*. More importantly, overexpression of RIDs using AAV9 was sufficient to repress pathological cardiac hypertrophy *in vivo*. These findings provided a potential gene therapy strategy for hypertrophic heart disease.

Taken together, we identified NCoR1 as a pivotal mediator, interacting with MEF2a and class IIa HDACs, to control the size of cardiomyocytes under both physiological and pathological conditions. These data unveiled novel functions of NCoR1 in cardiomyocytes and presented mechanistic insights. The findings support strategies targeting the NCoR1/MEF2a/HDACs complex in cardiomyocytes as potential novel approaches for the intervention of cardiac hypertrophy.

# Materials and Methods

## Animals and procedures

Cardiomyocyte-specific NCoR1 knockout (CMNKO) mice were generated by crossing floxed NCoR1 mice (Yamamoto *et al*, 2011) with αMHC-Cre mice (Duan *et al*, 2005). All mice were in C57BL6/J background. Abdominal aortic constriction (AAC) was performed as previously described (Hara *et al*, 2002; Li *et al*, 2017). Briefly, 8- to 12-week-old male mice were anesthetized with 2% isoflurane inhalation. Abdominal aortas were dissected and ligated against a blunted 27G needle using 6-0 silk sutures, and the needle was then removed. For sham operation, sutures were passed through without ligation. For angiotensin II (AngII)-induced cardiac hypertrophy, mini-pumps (2004D, Alzet) were implanted subcutaneously in 11- to 12-week-old male mice to deliver vehicle (sodium chloride) or AngII (750 ng/kg/min) for 4 weeks. Systolic and diastolic blood pressure was measured using a BP-2000 Blood Pressure Analysis System (Visitech Systems; Krege *et al*, 1995; Sun *et al*, 2017). For starvation experiments, mice were starved in the morning and sacrificed 24 h later. Sample size for all animal studies was determined using power analysis. Animals were excluded if not in good health condition. Animals with the same genotype were randomly allocated into different groups. Investigators assessing results of animal studies were blinded from group allocation.

Mice were housed no more than 5 per cage in a specific pathogen-free (SPF) facility under 12:12-h light–dark cycle, fed with standard rodent chow, and given sterilized drinking water *ad libitum*. All mouse experiments were carried out following the NIH Guide for the Care and Use of Laboratory Animals. All animal studies were approved by the Institutional Review and Ethics Board of Shanghai Ninth People's Hospital, Shanghai Jiao Tong University School of Medicine and the Institutional Animal Care and Use Committee of Institute for Nutritional Sciences, Shanghai Institutes for Biological Sciences, and Chinese Academy of Sciences.

## Human sample collection

Human heart samples were collected when patients with hypertrophic cardiomyopathy were undergone heart transplantation. Control heart samples were obtained from healthy donors with normal cardiac functions. Informed consent was obtained from all subjects, and all the experiments using human samples conformed to the principles set out in the WMA Declaration of Helsinki and the Department of Health and Human Services Belmont Report. The specimens were processed immediately after surgery, snap-frozen in liquid nitrogen, and then stored in a −80°C freezer. All studies were approved by the Institutional Review and Ethics Board of Shanghai Ninth People's Hospital, Shanghai Jiao Tong University School of Medicine.

## Cardiac hypertrophy estimation and sample collection

Mice were sacrificed 2 weeks after AAC. Ventricular weight-to-body weight ratio (VW/BW, mg/g) and ventricular weight-to-tibia length ratio (VW/TL, mg/mm) were used to evaluate cardiac hypertrophy. Ventricles were dissected, and parts of ventricles close to the base of the heart were fixed in 4% paraformaldehyde for histological

analysis. Left ventricles were separated and frozen in liquid nitrogen for further analyses. Left ventricles were separated and frozen in liquid nitrogen for further analyses. For RNA-sequencing, frozen ventricles were preserved in RNAlater (AM7030, Thermo Fisher Scientific).

### Echocardiograph analysis

Cardiac function was monitored using echocardiography (Vevo2100 Imaging System), similar to previously reported (Li *et al*, 2017). Mice were anesthetized, and hearts were imaged in two-dimensional LV long-axis view. Ejection fraction and fractional shorting were calculated based on M-mode recordings.

### Histological analysis

Paraformaldehyde-fixed heart samples were embedded in paraffin, and sections were stained with hematoxylin and eosin (H&E), WGA (W32464, Thermo Fisher Scientific), and 0.1% picrosirius red. Cross-sectional areas of cardiomyocyte were measured as previously described (Li *et al*, 2014; Yu *et al*, 2016). Fibrotic staining was quantified as a percentage of positively stained areas to the total areas examined.

### Triglyceride measurement and oil red O staining

For triglyceride measurement, lipids were extracted from ventricle samples using the Bligh/Dyer method (Bligh & Dyer, 1959). Briefly, ventricle samples were homogenized in 200 μl NaCl solution (0.1 M), and then, 1 ml chloroform/methanol (2:1) solution was added to each sample. The samples were kept shaking for 2 h at room temperature. After centrifugation at 3,000 *g* for 10 min, the chloroform layer was extracted and evaporated, and the resultant residue was resolved in 250 μl isopropanol. Triglycerides were measured using a quantification kit (Beijing BHKT Clinical Reagent). Oil red O staining was carried out following a previous report (Sun *et al*, 2011). In brief, frozen sections were fixed in formalin for 10 min, stained in 0.5% oil red O (CI 26125, Sigma-Aldrich) solution for 15 min, and then in hematoxylin for 5 s. Finally, sections were mounted in aqueous mounting media.

### Analysis of gene expression

Total RNA was isolated using TRIzol (15596026, Thermo Fisher Scientific), and cDNA was synthesized using reverse transcription kits (RR037A, Takara). qRT–PCR was carried out on an ABI7900HI (Applied Biosystems), and SYBR green (Applied Biosystems) was used to detect PCR products. Relative expression of each gene was determined by normalizing to GAPDH for ventricular samples and GAPDH or 18s for cells. The sequences of primers are listed in Appendix Table S5.

### Western blotting analysis

Total proteins were extracted by RIPA buffer containing protease inhibitor as previously reported (Sun *et al*, 2016; Zhang *et al*, 2017). Samples were separated by SDS–PAGE and then transferred to PVDF membranes. After primary and secondary antibody incubation, protein

expression was visualized using ECL Western Blotting Substrate (Thermo Fisher Scientific) and quantified using ImageJ software. The primary antibodies were anti-Actin (1:2,000, A7811, Santa Cruz), anti-α-Actinin (1:1,000, A7811, Sigma-Aldrich), anti-Flag (1:1,000, F3165, Sigma-Aldrich), anti-GFP (1:1,000, A10262, Thermo Fisher Scientific), anti-HA (1:1,000, H3663, Sigma-Aldrich or 3724, Cell Signaling), anti-HDAC9 (1:200, sc-398003, Santa Cruz; Schroeder *et al*, 2018), anti-MEF2a (1:200, sc-17785, Santa Cruz; Reineke *et al*, 2014), anti-NCoR1 (1:500, 5948, Cell Signaling; Simcox *et al*, 2015), and anti-HDAC4 (1:500, ab12172, Abcam or 2072, Cell Signaling; Wein *et al*, 2016).

### Chromatin immunoprecipitation (ChIP) and Co-immunoprecipitation (Co-IP)

Ventricular samples were homogenized into powders in mortars that were kept cold by lipid nitrogen. Then, the samples were crosslinked using 1% formalin solution for 15 min at room temperature, followed by quenching with 10X Glycine. ChIP experiments were carried out using EZ-ChIP kit (17-371, Millipore). 5 μg anti-NCoR1 antibodies (5948, Cell Signaling), 2 μg anti-MEF2a antibodies (sc-17785, Santa Cruz), 10 μg anti-HDAC4 (ab12172, Abcam), 10 μg anti-HDAC5 (ab1439, Abcam; Meng *et al*, 2017), 5 μg anti-acetyl-Histone H3 (06-599, Millipore), 5 μg anti-acetyl-Histone H4 (06-598, Millipore), 10 μg anti-RNA polymerase II (05-623B, Millipore), mouse IgG (12-371B, Millipore), and rabbit IgG (2729, Cell Signaling) were used for the pulldowns. ChIP products were analyzed using PCR. The PCR primers were as follows: *Acta1 F1* CAGCGGTA TAAATAGAACCCC, *Acta1 R1* CCTCCCAGGCAGACTCATCT and *Nppa F1* CATCCTGTTGGCACCTTGGACAC, *Nppa R1* CGTGTAAACACCA AGGGATG for ChIP using antibodies against NCoR1, anti-MEF2a, or anti-HDAC4; *Acta1 F2* CAGGCTGAGAACCAGCCGAAGGAA, *Acta1 R2* CCTGTCCCCTTGCACAGGTT and *Nppa F2* CGTGACAAGCTT TGCCGAAC, *Nppa R2* ATTCTGTCACTTGCAGCGATAAAG for ChIP using antibodies acetyl-Histone H3 or acetyl-Histone H4; and *Acta1 F3* GAAGACGAGACCACCGCTCTTG, *Acta1 R3* ATGGATGGGAACA CAGCCCTGG, *Nppa F3* TCCATCACCCTGGGCTTCTTCCT, and *Nppa R3* CCTTGAAATCCATCAGATCTGTG for ChIP using RNA polymerase II antibodies.

Co-IP experiments were performed as described previously (Sun *et al*, 2017). Ventricular samples were crashed and lysed by IP lysis buffer. The total lysates were separately incubated with anti-IgG (12-371B, Millipore) and anti-MEF2a antibodies (sc-17785, Santa Cruz) with protein A/G beads (sc-2003, Santa Cruz Biotechnology) overnight. Finally, the immunocomplex was subjected to Western blotting analysis.

### Cell culture and treatment

Neonatal rat ventricular myocytes (NRVMs) were isolated according to published protocols (Simpson & Savion, 1982; Wang *et al*, 2014). Briefly, ventricles were harvested from 1- to 2-day-old Sprague Dawley rat pups. After minced into small pieces, ventricles were digested in PBS containing 0.1% trypsin-EDTA and 1 mg/ml collagenase II (Worthington) to obtain single cell suspension. Cardiac fibroblasts were removed by preplating for 2–3 times. The plates were precoated with 1 mg/ml gelatin for 24 h. NRVMs were cultured in plating medium (high-glucose DMEM with 10% FBS, 1% penicillin–streptomycin, and 100 μM BrdU) for 24 h and

## The paper explained

### Problem

Heart failure is still a major cause of morbidity and mortality around the world. Pathological cardiac hypertrophy is an important risk factor for heart failure. Therefore, identifying effective targets and strategies to restrain pathological cardiac hypertrophy remains a promising approach to limit the rising incidence of heart failure. Nuclear receptor corepressor 1 (NCoR1) is an essential player in transcriptional regulation. It is unclear whether NCoR1 plays a role in cardiomyocytes and during cardiac hypertrophy under physiological and pathological conditions.

### Results

In this study, we found that cardiomyocyte-specific NCoR1 knockout (CMNKO) mice exhibited spontaneous cardiac hypertrophy. Compared to their control littermates, CMNKO mice had increased cardiac hypertrophy, heart failure, and early death upon pressure overload. Mechanistically, NCoR1 cooperated with class IIa histone deacetylases (HDACs) to repress transcriptional activities of myocyte enhancer factor 2 (MEF2). Importantly, we identified specific domains in NCoR1, the receptor interaction domains (RIDs), responsible for the effects of NCoR1 on MEF2. Overexpression of RIDs in myocardium attenuated pressure overload-induced cardiac hypertrophy and dysfunction.

### Impact

Our study identified NCoR1/MEF2/HDACs complex as a novel target in cardiac hypertrophy. Delivering RIDs of NCoR1 into myocardium is a promising therapeutic strategy to treat cardiac hypertrophy and heart failure.

then in maintaining medium (high-glucose DMEM with 1% penicillin–streptomycin and 100 μM BrdU). NRVMs were randomly divided into different groups. To induce cardiomyocyte hypertrophy, phenylephrine (PE, Sigma-Aldrich, 200 μM) was added into media after NRVMs were cultured in maintaining medium for 24–48 h. NRVMs were infected with adenoviruses containing HDAC4 (1435, Vector Biolabs), HDAC4-GFP (000548A, Applied Biological Materials Inc.), HDAC5-GFP (ADV-210890, Vector Biolabs), or HDAC9 (ADV-210895, Vector Biolabs) 24 h later after plating. 293 FT cells were from Cell Bank of Chinese Academy of Sciences and cultured in DMEM containing 10% FBS. The cell line was authenticated using short tandem repeat profiling and tested for mycoplasma, and no contamination was detected.

## Immunofluorescence staining

NRVMs cultured on glass coverslips were fixed in paraformaldehyde for 15 min and then washed for three times with 1XPBS. After incubation with blocking solution (5% goat serum and 0.3% Triton X-100 in 1X PBS) for 1 h, NRVMs were stained with anti-actinin (1:500, A7811, Sigma-Aldrich) overnight. Subsequently, NRVMs were incubated with secondary antibodies (1:1,000, A11005 or A11031, Thermo Fisher Scientific). Coverslips were finally mounted with ProLong® Gold Antifade Reagent with DAPI (P36931, Thermo Fisher Scientific). Images were captured using fluorescence microscope. Surface areas of NRVMs were quantified by ImageJ software. Cytosolic localization of HDAC4 and HDAC5 was calculated and presented as percentage (Toth et al, 2018).

## Plasmids and siRNA

NCoR1-flag plasmid was constructed as previously described (Yamamoto et al, 2011). Different fragments of NCoR1 were cloned into pHAGE plasmid. MEF2a-HA sequence was amplified from mouse cDNA and inserted into pcDNA 3.1. The sequences for siRNA were as follows: siNCoR1, 5′CCAGGUCCAUGACAAGUGA3′; siMEF2a, 5′CAGUCGGAAACCAGAUCUA3′; siMEF2c, 5′CUGGCAGCAAGAACACAAU3′; siMEF2d, 5′CACCAAGUUUACUCAGCCA3′ and siControl 5′UUCUCCGAACGUGUCACGUTT3′. NRVMs were transfected with non-targeting negative control and specific siRNA using Lipofectamine 2000 (Thermo Fisher Scientific) according to the manufacturer's instructions.

## Lentivirus packaging and infection

pHAGE-NCoR1-flag plasmid and empty pHAGE plasmid were introduced into 293FT cells together with lentivirus packaging plasmids using Lipofectamine 2000 (Thermo Fisher Scientific). Media containing lentivirus were collected 48 and 72 h after transfections, combined, filtered through 0.22 filters, and condensed by ultracentrifugation (Sun et al, 2016). NRVMs were infected by lentivirus 12 h later after plating.

## Production and delivery of adeno-associated virus serotype 9 (AAV9)

An AAV9 vector that contains cardiac troponin T (cTnT) promoter was constructed to make AAV9-cTnT-NCoR1 (1939-2453) plasmid by Hanbio Company (Shanghai, China). AAV-293 cells (Hanbio Company) were transfected with the AAV9-cTnT-NCoR1 (1939-2453) plasmids to produce virus. Viral particles were collected, purified, and concentrated. The titer was measured by qPCR. AAV9 was subcutaneously injected into neonatal mice as previously described (Pacak et al, 2006; Yu et al, 2016).

## Luciferase assay

For luciferase assay, a 2 kb fragment in mouse Acta1 or Nppa promoter region was cloned into pGL3 basic plasmid. Then, 293FT cells were co-transfected with the promoter constructs and NCoR1-flag, MEF2a-HA, and Renila plasmids separately or in combination. Transcriptional activities were measured using a Dual-Luciferase® Reporter Assay System (E1910, Promega) 48 h after transfections.

## RNA-sequencing

Frozen ventricles preserved in RNAlater were sent to WuXi NextCODE (Shanghai, China) for sequencing. Total RNA was extracted by TRIzol reagent (Thermo Fisher Scientific). After checking RNA quality, a sequencing library was constructed using TruSeq technology. Sequencing was carried out on a HiSeq X Ten (Illumina). Clean reads were aligned to mouse genome.

## Statistical analysis

Results were presented in the form of mean ± SEM and analyzed using Prism (GraphPad Software). Normality tests were performed to ensure normal distribution of data. Variation was detected for

each group of data, and similar variance was assured between statistically compared groups. Two-way ANOVA followed by Bonferroni post-tests was used for multiple comparisons. Student's *t*-test was used for pairwise comparisons. Results were considered significantly different if *P* values were ≤ 0.05.

## Data availability

Original data of RNA-sequencing were deposited into Gene Expression Omnibus (accession number GSE134923, https://www.ncbi.nlm.nih.gov/geo/query/acc.cgi?acc = GSE134923).

**Expanded View** for this article is available online.

## Acknowledgements
This work was supported by grants from the National Natural Science Foundation of China (81725003, 31671181, 91739303), the Ecole Polytechnique Fédérale de Lausanne (EPFL), and the Swiss National Science Foundation (31003A-140780). This work was also supported by the innovative research team of high-level local universities in Shanghai (oral and maxillofacial regeneration and functional restoration).

## Author contributions
CL, X-NS, B-YC, M-RZ, L-JD, TL, H-HG, Yuan L, Y-LL, L-JZ, X-JZ, Y-YZ, W-CZ, Yan L, CS, SS, X-RS, and YY performed experiments and collected data; CL, X-NS, B-YC, M-RZ, FJ, R-GL, and S-ZD designed experiments and analyzed data; XL and JW coordinated the study; JA provided NCoR1-floxed mice and revised the manuscript; CL, ZVW, and S-ZD wrote and revised the manuscript; FJ, R-GL, and S-ZD supervised and coordinated the study.

## Conflict of interest
The authors declare that they have no conflict of interest.

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
