## [Review Process File · EMBO Molecular Medicine]

Nuclear Receptor Corepressor 1 Represses Cardiac Hypertrophy

Chao Li, Xue-Nan Sun, Bo-Yan Chen, Meng-Ru Zeng, Lin-Juan Du, Ting Liu, Hui-Hui Gu, Yuan Liu, Yu-Lin Li, Lu-Jun Zhou, Xiao-Jun Zheng, Yu-Yao Zhang, Wu-Chang Zhang, Yan Liu, Chaoji Shi, Shuai Shao, Xue-Rui Shi, Yi Yi, Xu Liu, Jun Wang, Johan Auwerx, Zhao V. Wang, Feng Jia, Ruo-Gu Li, Sheng-Zhong Duan

Review timeline:

Submission date:	13th Mar 2018
Editorial Decision:	19th Mar 2018
New submission:	12th Mar 2019
Editorial Decision:	24th Apr 2019
Revision received:	24th Jul 2019
Editorial Decision:	13th Aug 2019
Revision received:	24th Aug 2019
Accepted:	27th Aug 2019

Editor: Lise Roth

Transaction Report:

1st Editorial Decision

19th Mar 2018

Thank you for submitting your manuscript to EMBO Molecular Medicine. I have now had a chance to read your research article carefully and to discuss it with the other members of our editorial team. I am sorry to inform you that we find that the manuscript is not well suited for publication in EMBO Molecular Medicine and that we therefore have decided not to proceed with peer review.

Your study investigates the role of the nuclear receptor corepressor 1 (NCoR1) in cardiomyocytes. Cardiomyocyte-specific deletion of NCoR1 in mice led to cardiac hypertrophy and impaired cardiac functions. Your data further suggest that NCoR1 inhibited cardiomyocytes hypertrophy through directly repressing the activities of MEF2.

We appreciate that your study uncovers new function for NCoR1 in cardiomyocytes. However, the *in vivo* translational applications of your work are not further developed here, thereby limiting the overall translational and clinical insights that are key for publication in EMBO Molecular Medicine. Therefore, I am afraid that we cannot offer further consideration to your article.

New submission - authors' response

12th Mar 2019

Last year, we submitted a previous version of this work to your journal. That version was rejected and the major concern of the editors was “...*the in vivo translational applications of your work are not further developed here, thereby limiting the overall translational and clinical insights that are key for publication in EMBO Molecular Medicine.*”

In response to this critique, we performed substantial additional experiments to support the translational and clinical potential of NCoR1 in cardiac hypertrophy. We mapped out that the C-terminal fragment of NCoR1 (also called RIDs) of NCoR1 was responsible for the inhibitory effects

on cardiac hypertrophy. Importantly, we delivered RIDs into myocardium using AAV9, and found it significantly repressed cardiac hypertrophy induced by pressure overload. These findings showed a great potential of gene therapy using RIDs of NCoR1 for hypertrophic heart disease. We also performed more mechanistic studies.

Our data have identified NCoR1 as a pivotal mediator to integrate the functions of MEF2a and HDAC4 during cardiac hypertrophy. The knowledge basis provided by this study may lead to effective new therapeutic strategies for pathological cardiac hypertrophy.

We believe that the manuscript has been significantly improved after the revision. Your favorable consideration of this revised version is highly appreciated.

2nd Editorial Decision

24th Apr 2019

Thank you for the submission of your manuscript to EMBO Molecular Medicine, and my apologies for the delay in getting back to you, which is due to the fact that one referee needed more time to complete his/her report.

We have now heard back from the 3 referees whom we asked to evaluate your manuscript, and as you will see from the reports below, they all acknowledge the potential interest and translational relevance of the findings, however they also have fundamental concerns that should be addressed in a major round of revision of the present manuscript.

Addressing the reviewers' concerns in full will be necessary for further considering the manuscript in our journal. EMBO Molecular Medicine encourages a single round of revision only and therefore, acceptance or rejection of the manuscript will depend on the completeness of your responses included in the next, final version of the manuscript.

REFeree REPORTS:

Referee #1 (Remarks for Author):

Li et al. Investigated the role of the nuclear receptor corepressor 1 (NCoR1) on cardiomyocyte hypertrophy, which has so far not been investigated. Using neonatal rat ventricular cardiomyocytes, the authors demonstrated that siRNA mediated knock-down leads to increased cardiomyocyte hypertrophy (with increased cell size and enhanced re-expression of embryonic genes, Acta1 and Nppa) after pro-hypertrophic stimulation with phenylephrine (PE), while lentiviral NCoR1 expression mediated the opposite, i.e. triggered anti-hypertrophic effects. For in vivo analyses, the authors generated cardiomyocyte specific NCoR1 knock-out mice (CM-NCoR1 KO), by crossing cardiomyocyte specific alpha-MHC-Cre mice with NCoR1 flox/flox mice. CM-NCoR1 KO mice exerted enhanced cardiomyocyte hypertrophy and cardiac dysfunction at the age of 10 months. Application of pressure-overload through abdominal aortic constriction at two months of age also triggered aggravated cardiac hypertrophy, fibrosis, dysfunction and increased mortality in the CM-NCoR1 KO mice. Mechanistically, the authors investigated the effects of NCoR1 on the pro-hypertrophic transcription factor MEF2A and MEF2D, since functional interaction between NCoR1 and MEF2 transcription factor was previously demonstrated in skeletal muscle cells (Reference #26 in the current manuscript). Li et al. demonstrate (via co-IP) that MEF2a and NCoR1 interact after their co-transfection in HEK293 cells and in mouse heart samples. The interaction domain is mapped in both proteins and it is shown that NCoR1 overexpression inhibits the activation of the ACTA1 and Nppa promoter. In turn, downregulation of Mef2a or Mef2d completely rescued enhanced hypertrophy of neonatal rat cardiomyocytes after knock-down of NCoR1, and ChIP assays showed enhanced MEF2a and reduced HDAC4 binding (going along with enhanced histone acetylation) in mouse hearts of CM-NCoR1 KO mice. Finally, the authors overexpressed the MEF2 binding fragment via AAV9-vector in the mouse myocardium and this led to reduced cardiac hypertrophy and improved heart function 2 weeks after abdominal aortic constriction. This is a potentially interesting, well conducted study that shows for the first time an antihypertrophic role of NCoR1 in the heart. This might be of clinical relevance in the future. On the

other hand, inhibition of MEF2 by NCoR1 had been previously demonstrated in skeletal muscle cells, thus limiting the mechanistic novelty of the author's findings. In addition, some important control groups are missing in some experiments and the expression/regulation of endogenous NcoR1 in cardiac disease is not reported. The authors have to address the following points:

Major

- 1) How is endogenous NCoR1 mRNA and/or protein expression regulated during hypertrophic stimulation in neonatal rat cardiomyocytes or after abdominal aortic constriction in mice? If NCoR1 expression is not altered, the authors should perform NCoR1 ChIP assay under hypertrophic stimulation to examine whether its DNA binding to the Acta1 or Nppa promoter is altered.
- 2) Similarly, the authors should examine NCoR1 expression or activity in human failing hearts versus healthy myocardium.
- 3) Figure 1A: Downregulation of NCoR1 by siRNA should also be demonstrated at the protein level. How many replicates are investigated in Figure 1 for the different panels?
- 4) Figure 2 and 3: It is known that the alpha-MHC-Cre mice develop cardiomyopathy with advanced age due to Cre toxicity (between 8-12 month of age, see Davis J. et al., *Circ. Res.*, 111: 761-777, 2012). Therefore, it is important to include an additional control group, in which alpha-MHC Cre is expressed on a wild-type background.
- 5) Echocardiographic parameter are reported very incompletely. Left ventricular enddiastolic diameters (or areas), as well as average wall thickness and heart rate have to be reported for each experiment. In addition, serial echocardiographic measurements are usually better compared to only one measurement/time point. In this regard, for Figure 7, a longer time point (e.g. 4 or 6 weeks after AAC) would be nice.
- 6) Online Figure S7: At least a group of AAC mice without MC1568 needs to be added to all panels. If this is not possible, this data should be removed. It is inconclusive as it stands.
- 7) Is HDAC4 nuclear localization affected by the manipulation of NCoR1 in NRVMs?

Minor:

- 8) N-numbers need to be consistently shown for each panel.
- 9) Figure 7G: The fibrotic area needs to be quantified and statistical analysis needs to be performed.
- 10) Online Figure S5: Cardiomyocyte size needs to be quantified. From the pictures it looks like siMEF2C does reduce hypertrophy during siNCoR1 treatment.
- 11) The English needs to be improved and edited in some passages:
e.g. "we postulated whether NCoR1 would affect lipid metabolism in the heart" better would be: "we analyzed whether..."; "Echocardiograph analysis" instead of "Echocardiographic analysis".
"MEF2 acetylation is associated with its transcriptional activity and its interaction with HDACs" this sentence needs to be re-written and better explained.
- 12) Page numbers need to be introduced.

Referee #2 (Remarks for Author):

This is an interesting study that focuses on the role of cardiac-expressed Ncor1 in cardiac hypertrophy. The data are novel and potentially of high interest. But I have 3 major points:

1. Data derived from a genetic mouse model are convincing but are for my feeling reach not far enough. I would like to see not only cardiac hypertrophy (and a bit of cardiac function) as major readout but also molecular/cellular changes that explain the dysfunctional myocardium. Can e.g. MEF2-dependent processes like inflammation or glucose metabolites be detected? This can be descriptive but would give more confidence to judge the contribution of MEF2 to the disease phenotype.
2. The mechanist data are interesting, but also a bit predictable from the literature and also for my taste a bit too premature. What about the other class II HDACs that can bind Ncor1. HDAC4 is not really involved in hypertrophy but rather metabolic remodeling. HDAC5 and HDAC9 control cardiac hypertrophy. Do they also bind to Ncor1? What is the upstream activator? Phenylephrine was used before bit not in these experiments. HDAC4 responds to PKD and CaMKII. HDAC5 and 9 not to CaMKII. Can the authors define the mechanism a bit deeper to come to a better understanding. Binding experiments would be good to see as well.
3. An unbiased approach to define more specific Ncor1 target genes would be very helpful as well to see whether there is a partial or complete similar basis of Ncor1 and MEF2.

Referee #3 (Remarks for Author):

This manuscript by Li et al. is well written paper and describes the roles of NCoR1, MEF2 and HDAC4 in the formation of cardiac hypertrophy. The readers will benefit more from reading this paper with several additional explanations.

Major comments

The interaction between NCoR1 and MEF2a is only shown by overexpression of these genes (Figure 5). The direct association of these endogenous proteins should be shown.

1st Revision - authors' response

24th Jul 2019

Response to Referee #1

Major

1) How is endogenous NCoR1 mRNA and/or protein expression regulated during hypertrophic stimulation in neonatal rat cardiomyocytes or after abdominal aortic constriction in mice? If NCoR1 expression is not altered, the authors should perform NCoR1 ChIP assay under hypertrophic stimulation to examine whether its DNA binding to the Acta1 or Nppa promoter is altered.

Response: Following the reviewer's suggestion, we measured protein expression of NCoR1 in neonatal rat cardiomyocytes after phenylephrine treatment and in mouse heart samples after abdominal aortic constriction. The results showed that protein expression of NCoR1 significantly increased after hypertrophic stimulation (**Figure EV1H-J**). These results suggested that NCoR1 was stress-responsive and part of the adaptation program during cardiac hypertrophy. Multiple similar molecules (with increased expression responding to hypertrophic stimuli and yet suppressed cardiac hypertrophy) have been reported previously. We have discussed this in details in the Discussion section.

2) Similarly, the authors should examine NCoR1 expression or activity in human failing hearts versus healthy myocardium.

Response: We collected heart samples from health donors and hypertrophic cardiomyopathy (HCM) patients. Firstly, we evaluated the quality of these samples by examining the expressions of hypertrophic genes. As expected, expression of *Nppa*, *Nppb* and *Myh7* was all increased in heart samples from HCM patients (**Figure EV1A**). Then we measured protein levels and found that the expression of NCoR1 was also elevated in hypertrophied human hearts (**Figure EV1B-C**).

3) Figure 1A: Downregulation of NCoR1 by siRNA should also be demonstrated at the protein level. How many replicates are investigated in Figure 1 for the different panels?

Response: Following the reviewer's suggestion, we replaced the qPCR results with Western blotting results, which demonstrated the decreased protein expression of NCoR1 in neonatal rat cardiomyocytes (**Figure 3A**).

Each panel contains 3 or 4 replicates in one experiment. A total of 30-40 cardiomyocytes was randomly chosen from these 3-4 replicates for quantification. We indicated them in the revised figure legends for Figure 2 and other related figures as well.

4) Figure 2 and 3: It is known that the alpha-MHC-Cre mice develop cardiomyopathy with advanced age due to Cre toxicity (between 8-12 month of age, see Davis J. et al., *Circ. Res.*, 111: 761-777, 2012). Therefore, it is important to include an additional control group, in which alpha-MHC Cre is expressed on a wild-type background.

Response: To evaluate the possible Cre toxicity to the heart, we obtained an age-matched cohort of α MHC-Cre mice on wild-type background without any loxP sites and compared them to LC (floxed NCoR1) mice and CMNKO mice. At the age of 10 months, cardiac function was comparable

between α MHC-Cre mice and LC mice (**Figure EV2A and Appendix table S2**), suggesting no cardiac toxicity of Cre in this line of α MHC-Cre mice.

Different lines of α MHC-Cre mice may explain the discrepancy. The line used in our study was originally reported in *J Clin Invest.* 1999 Dec 15; 104(12): 1703–1714. The one mentioned by the reviewer was originally reported in *J Clin Invest.* 1997 Jul 1;100(1):169-79. The two lines were generated using different approaches. In addition, there were other studies that reported no Cre toxicity in old α MHC-Cre mice (e.g. *Circulation.* 2019 Jun 10. doi: 10.1161/CIRCULATIONAHA.118.038924).

5) Echocardiographic parameter are reported very incompletely. Left ventricular enddiastolic diameters (or areas), as well as average wall thickness and heart rate have to be reported for each experiment. In addition, serial echocardiographic measurements are usually better compared to only one measurement/time point. In this regard, for Figure 7, a longer time point (e.g. 4 or 6 weeks after AAC) would be nice.

Response: Following the reviewer's suggestion, we included complete parameters of echocardiography for each experiment (**Appendix table S1-4**). For figure 7, a new cohort of mice was subjected to AAC and injected with AAV9 GFP or AAV9 NCoR1 (1939-2453). We detected cardiac function for these mice before surgery, 2 weeks and 4 weeks after surgery (**Figure 7J, K, Appendix table S4**).

6) Online Figure S7: At least a group of AAC mice without MC1568 needs to be added to all panels. If this is not possible, this data should be removed. It is inconclusive as it stands.

Response: We agree that it would be more convincing if the group of AAC mice without MC1568 was included in these results. However, we did not obtain enough mice for this additional experiment over the time period of revision. Following reviewer's suggestion we removed the data from the manuscript.

7) Is HDAC4 nuclear localization affected by the manipulation of NCoR1 in NRVMs?

Response: To address this question, we expressed fusion protein HDAC4-GFP with adenovirus in NRVMs and used GFP to indicate HDAC4 translocation. PE stimulation significantly increased translocation of HDAC4 from nucleus to cytosol and NCoR1 overexpression inhibited this process (**Figure EV5A, B**). In addition, as suggested by Referee #2 other class IIa HDACs may be involved in the repressive functions of NCoR1, we explored the impacts of NCoR1 on translocation of HDAC5 and revealed similar results (**Figure EV5C, D**).

Minor:

8) N-numbers need to be consistently shown for each panel.

Response: Following the reviewer's suggestion, we added n-numbers to those panels lacked or modified the format of some of the existing ones throughout the manuscript.

9) Figure 7G: The fibrotic area needs to be quantified and statistical analysis needs to be performed.

Response: We quantified the fibrosis area and performed the statistical analysis as shown in **Figure 7H**.

10) Online Figure S5: Cardiomyocyte size needs to be quantified. From the pictures it looks like siMEF2C does reduce hypertrophy during siNCoR1 treatment.

Response: We quantified the size of cardiomyocytes for this panel. The results showed that siMEF2c did not significantly reduce hypertrophy after siNcoR1 treatment (**Figure EV4C**).

11) The English needs to be improved and edited in some passages:

e.g. "we postulated whether NCoR1 would affect lipid metabolism in the heart" better would be: "we analyzed whether..."; "Echocardiograph analysis" instead of "Echocardiographic analysis". "MEF2 acetylation is associated with its transcriptional activity and its interaction with HDACs" this sentence needs to be re-written and better explained.

Response: We thoroughly checked the language and extensively revised the whole manuscript to improve readability.

12) Page numbers need to be introduced.

Response: We added page numbers to the manuscript.

Response to Referee #2

1. Data derived from a genetic mouse model are convincing but are for my feeling reach not far enough. I would like to see not only cardiac hypertrophy (and a bit of cardiac function) as major readout but also molecular/cellular changes that explain the dysfunctional myocardium. Can e.g. MEF2-dependent processes like inflammation or glucose metabolites be detected? This can be descriptive but would give more confidence to judge the contribution of MEF2 to the disease phenotype.

Response: Following the reviewer's suggestion, we measured expression of genes related to inflammation or glucose metabolism in left ventricle (LV) samples from LC and CMNKO mice. MEF2 has been shown to regulate expression of cJun (*Nature*. 1997 Mar 20;386(6622):296-9) and MCP1 (*Circ Res*. 2004 Jul 9;95(1):42-9). Our results showed that expression of inflammatory genes such as cJun and MCP1 was significantly increased in LV samples from CMNKO mice compared to those from LC mice (**Appendix Figure S2C**).

MEF2 has also been shown to regulate Glut4 (*Genes Dev*. 2016 Feb 15;30(4):434-46) as well as orphan nuclear receptors Nurr77/Nr4a1 and Nurr1/Nr4a2 (*Genes Dev*. 2016 Feb 15;30(4):434-46; *Nat Med*. 2018 Jan;24(1):62-72). Our results demonstrated that expression of Glut4 was mildly increased in LV samples from CMNKO mice (**Appendix Figure S2C**). Expression of Nr4a1 or Nr4a2 was not significantly different in LV samples between LC and CMNKO (**Figure for Reviewers 1**).

2. The mechanist data are interesting, but also a bit predictable from the literature and also for my taste a bit too premature. What about the other class II HDACs that can bind Ncor1. HDAC4 is not really involved in hypertrophy but rather metabolic remodeling. HDAC5 and HDAC9 control cardiac hypertrophy. Do they also bind to Ncor1? What is the upstream activator? Phenylephrine was used before bit not in these experiments. HDAC4 responds to PKD and CaMKII. HDAC5 and 9 not to CaMKII. Can the authors define the mechanism a bit deeper to come to a better understanding. Binding experiments would be good to see as well.

Response: We performed a series of additional experiments to address these questions. First, we performed ChIP experiments to detect recruitment of HDAC5 and HDAC9 on Acta1 and Nppa promoters in ventricular samples. The results showed that less HDAC5 were recruited to the MEF2 binding sites on Acta1 and Nppa promoters in ventricular samples from CMNKO mice compared to those from LC mice (Figure 6D), indicating the importance of HDAC5 in controlling cardiac hypertrophy. However, due to technical difficulty (lack of good antibodies), we were not able to obtain ChIP results for HDAC9.

Secondly, we tested whether NCoR1 directly interacted with HDAC4, HDAC5 or HDAC9 in cardiomyocytes. As shown in **Figure EV5E**, we did not detect any interactions between NCoR1 and these HDACs in neonatal rat ventricular myocytes (NRVMs). These cells were overexpressed with NCoR1 and HDACs but not MEF2a, suggesting that NCoR1 did not directly bind to either of these HDACs and that MEF2 might be a critical component bridging NCoR1 and HDACs. In fact, when MEF2a was overexpressed, interaction between NCoR1 and HDAC5 was observed in NRVMs (Figure EV5G), further suggesting the importance of MEF2 as a connecting point. Since we did not detect direct interactions between NCoR1 and class IIa HDACs, we did not go on to identify the upstream activators.

Thirdly, we further tested the ability of HDAC4, HDAC5 and HDAC9 to repress hypertrophy in cultured NRVMs. The results showed that overexpression of any of these 3 HDACs significantly decreased PE-induced hypertrophy in NRVMs (**Figure for Reviewers 2**). Although HDAC4 knockout mice manifest normal hypertrophic response after pressure over-load (*Nat Med*. 2018 Jan;24(1):62-72), the inhibitory effects on cardiomyocyte hypertrophy by HDAC4 overexpression in NRVMs has been reported before (*J Cell Biol*. 2011 Oct 31;195(3):403-15). We suspect that HDAC4 deficiency in vivo may have led to compensation by other class IIa HDACs such as HDAC5 and HDAC9, which may explain the lack of phenotype in HDAC4 knockout mice.

We sincerely thank the reviewer for these series of questions, which greatly extended our understanding about the interactions between NCoR1 and class IIa HDACs in cardiomyocytes. Based on the results related to these questions we revised our working model (**Figure 5H**).

3. An unbiased approach to define more specific Ncor1 target genes would be very helpful as well to see whether there is a partial or complete similar basis of Ncor1 and MEF2.

Response: We performed RNA-seq analysis of LV samples from LC and CMNKO mice (**Appendix Excel**). We paid particularly attention to the up-regulated genes in CMNKO mice, and found 57 genes were most significantly increased (Fold change>2, FDR<0.05) (**Appendix Excel, Appendix Figure S2A**). Among these 57 genes, 11 genes were reported to be regulated by MEF2 (**Appendix Excel, Appendix Figure S2B**). These results support that there is a partial similar basis of NCoR1 and MEF2 regarding gene regulation in the heart.

Response to Referee #3

Major comments

The interaction between NCoR1 and MEF2a is only shown by overexpression of these genes (Figure 5). The direct association of these endogenous proteins should be shown.

Response: The interactions of endogenous MEF2a, HDAC4 and NCoR1 in heart samples were examined by Co-IP using antibodies against MEF2a (**Figure EV5F**).

Figure for Reviewers 1

Figure for Reviewers 1. qRT-PCR analysis of expression of genes related to glucose metabolism in left ventricles. n=9:9:11:8. N.S., not significant.

3rd Editorial Decision

13th Aug 2019

Thank you for the submission of your revised manuscript to EMBO Molecular Medicine. We have now received the referees' reports, and as you will see they are supportive of publication of your study. I am therefore pleased to inform you that we will be able to accept your manuscript pending the following final editorial amendments:

REFeree REPORTS:

Referee #1 (Remarks for Author):

The authors adequately addressed my concerns. This is an interesting paper.

Referee #2 (Remarks for Author):

no comments left

Corresponding Author Name: Sheng-Zhong Duan

Journal Submitted to: EMBO molecular medicine

Manuscript Number: EMM-2018-09127-V3